# EDMF: A New Benchmark for Multi-Focus Images with the Challenge of Exposure Difference

**DOI:** 10.3390/s24227287

**Published:** 2024-11-14

**Authors:** Hui Li, Tianyu Shen, Zeyang Zhang, Xuefeng Zhu, Xiaoning Song

**Affiliations:** International Joint Laboratory on Artificial Intelligence of Jiangsu Province, School of Artificial Intelligence and Computer Science, Jiangnan University, Wuxi 214122, China; 6223114021@stu.jiangnan.edu.cn (T.S.); zzy_jnu_cv@163.com (Z.Z.); xuefeng.zhu@jiangnan.edu.cn (X.Z.); x.song@jiangnan.edu.cn (X.S.)

**Keywords:** dataset, multi-focus image fusion, exposure difference, memory unit

## Abstract

The goal of the multi-focus image fusion (MFIF) task is to merge images with different focus areas into a single clear image. In real world scenarios, in addition to varying focus attributes, there are also exposure differences between multi-source images, which is an important but often overlooked issue. To address this drawback and improve the development of the MFIF task, a new image fusion dataset is introduced called EDMF. Compared with the existing public MFIF datasets, it contains more images with exposure differences, which is more challenging and has a numerical advantage. Specifically, EDMF contains 1000 pairs of color images captured in real-world scenes, with some pairs exhibiting significant exposure difference. These images are captured using smartphones, encompassing diverse scenes and lighting conditions. Additionally, in this paper, a baseline method is also proposed, which is an improved version of memory unit-based unsupervised learning. By incorporating multiple adaptive memory units and spatial frequency information, the network is guided to focus on learning features from in-focus areas. This approach enables the network to effectively learn focus features during training, resulting in clear fused images that align with human visual perception. Experimental results demonstrate the effectiveness of the proposed method in handling exposure difference, achieving excellent fusion results in various complex scenes.

## 1. Introduction

As a highly significant branch of multi-sensor information fusion, image fusion technology [1,2,3] has attracted extensive attention and research worldwide in the past two decades. MFIF is a crucial technique [4,5,6] for producing globally clear images through the fusion of images captured at different focal depths. This method is widely applied in various domains, including medical imaging [7,8], satellite remote sensing [9], and smartphone photography [10,11].

Despite the vast applications of MFIF, it still faces several challenges. One of them is the exposure difference between multi-focus images. When capturing objects at different distances, images often exhibit varying degrees of exposure difference. The main cause of this issue is the automatic aperture adjustment of smartphones. Fusing multi-focus images with exposure difference into a visually pleasing and clear image is a problem that urgently needs to be addressed. It is worth noting that there are currently no publicly available multi-focus image datasets related to this issue.

The public MFIF datasets can be categorized into two types based on the generation methods: synthetic and captured. Synthetic datasets include MFI-WHU [12], BetaFusion [13], and WHU-MFM [14]. These datasets are commonly created by applying a Gaussian filter to segmented datasets with masks, which blurs them into foreground and background, thus constituting multi-focus image pairs. Unfortunately, synthetic datasets fail to reflect the characteristics of real complex scenes.

Captured datasets such as Lytro [15], MFFW [16], and Road-MF [17] offer a different perspective. The MFFW dataset presents relatively complex scenes. However, it contains only a single pair of multi-focus images with exposure variations. In contrast, the scenarios in Lytro and Road-MF are simpler and more idealized. Considering the current situation where there are no datasets that consider exposure difference, the EDMF has been developed. The EDMF dataset encompasses a wide variety of complex scenes and specifically includes multiple exposure variation phenomena. The EDMF has been made public and is licensed under CC BY 4.0.

Figure 1a provides a simple comparison between our dataset and other datasets. As shown in Figure 1b, compared with the public datasets, EDMF contains more images for scholars to research. In addition, the image resolution is not fixed. These points make EDMF more challenging. As shown in Figure 1c, existing deep learning-based MFIF methods [18] cannot effectively solve the exposure difference.

To address the issue and provide a baseline method, in this paper, an unsupervised model incorporating multiple memory unit structures is introduced. To the best of our knowledge, this is the first network architecture and dataset that consider the varying exposure levels in multi-focus images task according to real-world scenarios. The main contributions of this paper are summarized as follows:A novel multi-focus dataset with varying exposure levels is provided, which covers diverse scenarios such as indoor, outdoor, daytime, and nighttime environments.An improved unsupervised model is proposed to address this new issue; it is capable of merging images with different focus attributes and brightness levels into a clear image that aligns with human perception.During the training process, we devise multiple adaptive memory units and utilize spatial frequencies as pixel value weights for intermediate results, allowing the network to focus more on the focused regions.

Experimental results demonstrate that this design can generate fusion images with outstanding visual quality, meeting our expectations.

## 2. Related Work

### 2.1. Classification-Based MFIF Methods

These types of methods [23,24,25] fundamentally utilize network to extract features from the source images and perform binary classification on the pixel-wise focusing properties to obtain decision maps. The fusion image is obtained by multiplying two source images separately with a decision map and then adding them together.

In 2017, Liu et al. [26] first introduce convolutional neural networks as binary classification networks into MFIF. To improve the accuracy of decision maps, in 2021, Ma et al. [22] combine spatial frequencies from the traditional method with an autoencoder. In 2022, Kanika Bhalla et al. [27] proposed to combine the advantages of fuzzy sets (FS) and convolutional neural networks (CNN) to generate high-quality fused images, thus overcoming the limitations of traditional manual design of fusion rules and feature extraction. It is worth mentioning that this technique has also been applied in the fusion of infrared and visible images [28]. In the past two years, due to the lack of training datasets, many unsupervised methods have been proposed. In 2023, Xu et al. pioneered the application of a zero-shot method to the MFIF task [20], achieving outstanding fused image results. However, this approach incurred substantial time costs. In 2024, Wang et al. employed a non-deep learning approach [21], alleviating the reliance on extensive training data. They devised a fusion method based on multi-dictionary linear sparse representation, achieving state-of-the-art performance. It still relies on initially obtaining the focus decision map initially but faces challenges in generating accurate maps under complex scene scenarios.

Specifically, although this type of method can perfectly extract information from the source images, the strategy of the decision map is completely not applicable to complex scenarios with exposure differences. It is precisely this strategy that leads to the extraction of image patches with inconsistent luminance information from the two source images, thereby resulting in extremely poor visual effects of the fused image.

### 2.2. Regression-Based MFIF Methods

Unlike classification-based methods, regression-based methods [29,30] directly output fused images, constituting an end-to-end fusion approach. These works typically comprise three parts: feature extraction, feature fusion, and image reconstruction.

For example, in 2022, Xu et al. proposed U2Fusion [31], which employs a feature extractor with dense connections to obtain features from the source images and trains a reconstruction module to generate fused results. However, since this method is not specifically designed for MFIF, the fused image may still be blurry in some regions. In the same year, Cheng et al. propose MUFusion [32]. It incorporates neural networks with memory unit structures into image fusion, enhancing the ability of network to extract and fuse image features. It should be noted that the weight setting of the MUFusion loss function is not designed for MFIF.

Interestingly, regression-based methods such as U2Fusion and MUFusion are more robust to exposure difference than classification-based methods. This is because of this method independence from decision maps. When it comes to dealing with exposure differences, feature-level fusion is capable of generating results that are more in line with human vision. Although the problem of exposure differences has been solved, the existing regression-based methods, when dealing with multi-focus image tasks, especially in the design of loss functions, do not take into account the targeted design of focusing weights to improve the quality of the fused images. Therefore, there is still room for improvement in the clarity of the fused images obtained by such methods at present.

### 2.3. MFIF Dataset

A well-designed test dataset plays a crucial role in assessing the performance of algorithms. Currently, several publicly available datasets for multi-focus image fusion (MFIF) exist, including Lytro [15], MFFW [16], MFI-WHU [12], Road-MF [17] and BetaFusion [13].

As shown in Figure 1b, the differences between these datasets in scale and resolution have been introduced in the introduction. Moreover, Lytro, MFFW, and MFI-WHU are widely used by scholars at present.

The images in the Lytro dataset are captured by the Lytro camera. The scene is very simple. Therefore, it is easy to detect the focus map. With this in mind, the MFFW dataset has been proposed. The dataset comes from real images collected from the Internet. It has complex scenes, but the scale is small, and only one pair of images has an exposure difference. In order to expand the scale of tests, the MFI-WHU dataset has been proposed. It is synthesized and constructed based on the public datasets. Although the scale is expanded, the scenario is still too simple.

Recently, in order to combine MFIF with downstream tasks, some scholars have proposed the Road-MF. It has a single scene, including only roads and vehicles. In order to solve the transition region (TR) problem, some scholars have proposed the Betafusion dataset, which has challenging but only contains 80 pairs.

It is worth noting that there is no publicly available dataset that addresses exposure differences in MFIF. To this end, the EDMF dataset is proposed. It has three highlights, including exposure difference, a relatively large data volume, and a large number of complex scenes that are challenging. Table 1 summarizes the characteristics of the EDMF dataset and other datasets.

## 3. Dataset

### 3.1. Dataset Construction

#### 3.1.1. The Equipment and Process of Image Acquisition

Four different smartphone models are employed for dataset collection, namely, IQOO Z8, IQOO Neo8 Pro, Redmi K70, and iPhone 13, along with their respective operating systems (three on Android and one on iOS). To ensure the authenticity of dataset, no alterations are made to the camera settings during image capture, accurately representing real-world usage scenarios. Both tripods and smartphone mounts are employed during the image capture process. It minimizes camera shake and misalignment.

Scenes encompass diverse locations, including campuses, communities, roads, and both daytime and nighttime settings. In total, 1000 pairs of images were captured.

#### 3.1.2. Image Registration

Image registration, particularly for manually captured images, presents challenges due to inherent variations in camera positioning and orientation.

Thus, the Position, Scale, and Orientation-invariant Scale-Invariant Feature Transform (PSO-SIFT) [33] method is employed for image registration. This method falls under the category of feature point matching algorithms.

As illustrated in Figure 2, the registration process includes the following steps: feature point extraction, feature point matching, computation of transformation matrices, and resampling of the source image to obtain properly registered images.

In our dataset (EDMF), there are diverse scenes, including day and night, indoor and outdoor, as shown in the Figure 3. It is normal for the registered image to have black edges. This is because registration uses one image as a reference and the other image is aligned with it through translation and rotation operations.

## 4. Proposed Method

In this section, an enhanced unsupervised model centered around the memory unit structure is introduced. Initially, the motivation for enhancing the memory unit structure is discussed, and the design strategy for improving the model to solve the multi-focus image fusion problem in complex real-world scenes is presented in detail.

Following the above processing, a thorough introduction to the loss function customized for the MFIF task and its weight distribution is provided. This design focuses on the clarity and coherence of the fused image, ensuring that the model minimizes the impact of exposure inconsistencies while maintaining image clarity.

Finally, the optimization process of the network structure is presented, enhancing the efficiency of model and deployability through streamlined operations while preserving performance integrity.

### 4.1. Enhanced Memory Unit

In MUFusion [32], the memory unit structure plays a crucial role by utilizing the output of the previous training epoch to guide the fusion process in the current training epoch. This structure enables the retrieval of intermediate results generated during the training process, providing rich supervisory signals to guide the training of network. This training method relies on the design of intermediate results and the loss function.

However, as the training progresses, relying solely on the output of the previous epoch may not always be optimal due to the best output of one epoch may not necessarily effectively promote the training of subsequent epochs. Additionally, the optimal solution may already exist in some previous epochs.

To address this issue, as illustrated in Figure 4, an improved memory unit structure is proposed in this paper. In this structure, not only the previous epoch is considered, but it also integrates the first four best outputs from the previous training epochs to guide the current training epoch. The key aspect of this approach lies in how to select these four optimal training epochs.

Spatial frequency [34,35], as a common image processing operation, can reflect the sharpness of the image to some extent. Based on this observation, in our method, spatial frequency is used to evaluate and filter the quality of the outputs at each stage of the network. This filtering process is described in Algorithm 1.    
**Algorithm 1:** Screen out the top four epochs of fusion quality
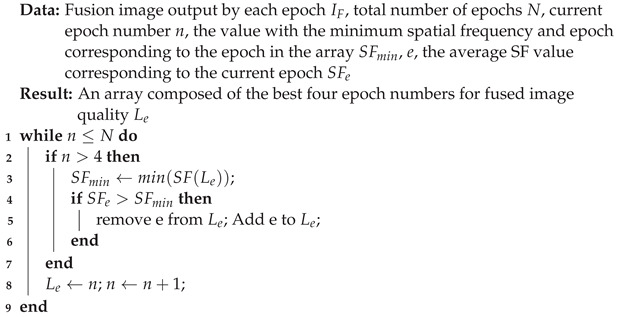


### 4.2. Loss Function

In MUFusion [32], the pre-trained VGG-19 network [36] is used to extract deep features from the two input source images. The mask is generated by directly comparing the magnitudes of the elements in the feature matrix and is used as the weight in the loss function to effectively preserve the prominent details. This enables the network to focus on the regions with rich details in the two original images during the training process, so that the fused image can retain as many texture details as possible from the original images.

However, the way of obtaining the mask weight in MUFusion is too simple and rough, and there is no specific design for the MFIF task, so it is unable to accurately distinguish the focused and unfocused regions.

To extract focus-related activation level maps in a more reasonable manner, this study employs a method based on spatial frequency. The obtained mask is only used as a weight matrix in loss function for the current epoch. Specifically, the spatial frequency between two source images is calculated firstly. It reflects the sharpness of different regions in the images. Subsequently, a more accurate mask is obtained by comparing these spatial frequencies at the pixel level. The obtained mask indicates which parts are sharply focused and which parts may be blurred.

Through the proposed mask extraction strategy, more attention is ensured to be paid by the network to the truly important and well-focused areas during the fusion process, thereby enhancing the quality and effectiveness of multi-focus image fusion. Specifically, this paper cites three kinds of loss, which are pixel loss, gradient loss, and structural similarity loss. The following is the specific loss function formula:(1)$$L_{pixel}^{i}={∥\sum_{x=1}^{M}\sum_{y=1}^{N}\left(I_{O}\left(x,y\right)-I_{O_{i}}\left(x,y\right)\right)∥}_{F}^{2},\;s.t.\;i\in \left\{1,2,3,4\right\}$$
(2)$$L_{pixel}^{\prime }={∥M_{1}\times \sum_{x=1}^{M}\sum_{y=1}^{N}\left(I_{O}\left(x,y\right)-I_{1}\left(x,y\right)\right)∥}_{F}^{2}+{∥M_{2}\times \sum_{x=1}^{M}\sum_{y=1}^{N}\left(I_{O}\left(x,y\right)-I_{2}\left(x,y\right)\right)∥}_{F}^{2}$$
(3)$$L_{grad}^{i}={∥\sum_{x=1}^{M}\sum_{y=1}^{N}▽I_{O}\left(x,y\right)-\sum_{x=1}^{M}\sum_{y=1}^{N}▽I_{O_{i}}\left(x,y\right)∥}_{F}^{2},\;s.t.\;i\in \left\{1,2,3,4\right\}$$
(4)$$L_{grad}^{\prime }={∥M_{1}\times \sum_{x=1}^{M}\sum_{y=1}^{N}▽\left(I_{O}\left(x,y\right)-I_{1}\left(x,y\right)\right)∥}_{F}^{2}+{∥M_{2}\times \sum_{x=1}^{M}\sum_{y=1}^{N}▽\left(I_{O}\left(x,y\right)-I_{2}\left(x,y\right)\right)∥}_{F}^{2}$$
(5)$$L_{ssim}^{i}=\left(1-SSIM\left(I_{O},I_{O_{i}}\right)\right),\;s.t.\;i\in \left\{1,2,3,4\right\}$$
(6)$$L_{ssim}^{\prime }=2-SSIM\left(M_{1}\times I_{O},M_{1}\times I_{1}\right)+SSIM\left(M_{2}\times I_{O},M_{2}\times I_{2}\right)$$

In above formulas, × indicates element-wise multiplication. ${∥\cdot ∥}_{F}$ indicates the Frobenius norm. *M* and *N* are the dimensions of the image in terms of row and column, respectively. $I_{1}$ and $I_{2}$ represent the two source images. (x,y) represents the pixel value. $M_{1}$ and $M_{2}$ represent the mask of the focus area obtained from the source image using the spatial frequency. $L_{pixel}^{\prime }$, $L_{grad}^{\prime }$, and $L_{ssim}^{\prime }$ represent the losses calculated in the current epoch. $L_{pixel}^{i}$, $L_{grad}^{i}$, and $L_{ssim}^{i}$ calculate the loss value between the output $I_{O}$ of the current epoch and the output $I_{O_{i}}$ of the four best epochs in the previous epochs. Based on the above, the loss value of the current epoch can be obtained as follows:(7)$$L_{current}=L_{pixel}^{\prime }+L_{grad}^{\prime }+L_{ssim}^{\prime }$$

The output in the previous epoch and the output result of the current epoch are utilized to calculate the loss value as memory loss:(8)$$L_{memory}^{i}=L_{pixel}^{i}+L_{grad}^{i}+L_{ssim}^{i},\;s.t.\;i\in \left\{1,2,3,4\right\}$$

In order to achieve effective training supervision, an adaptive loss ratio is also introduced, which can be adjusted adaptively according to the characteristics of different training epochs. It is primarily based on the Structural Similarity Index (SSIM) which is utilized to calculate the weights of the loss functions.

Thus, the loss ratio can be flexibly adjusted to fully utilize the intermediate fusion results, thereby improving the quality of the final fused image. The calculation process of adaptive ratio between loss functions is as follows:(9)$$S=[SSIM\left(I_{O},I_{1}\right)+SSIM\left(I_{O},I_{2}\right)]/2$$
(10)$$S_{i}=\left[SSIM\left(I_{O_{i}},I_{1}\right)+SSIM\left(I_{O_{i}},I_{2}\right)\right]/2,\;s.t.\;i\in \left\{1,2,3,4\right\}$$
(11)$$W=\frac{e^{S}}{e^{S}+\sum_{i=1}^{4}e^{S_{i}}}$$
(12)$$W_{i}=\frac{e^{S_{i}}}{e^{S}+\sum_{i=1}^{4}e^{S_{i}}}$$

In the above equations, SSIM(·) denotes the Structural Similarity Index between two images. *W* and $W_{i}$ are used to control the preservation degrees of the current output and the previous outputs in the loss function.

Based on the current loss and the memory loss, the total loss is given as follows:(13)$$L_{total}=W\times L_{current}+\sum_{i=1}^{4}W_{i}\times L_{memory}^{i}$$

After the network is improved through enhancing the training strategy and precisely designing the weight of the loss function, it is capable to adaptively fuse the areas in the source images that are rich in detailed textures. Consequently, satisfactory fusion results can be generated.

### 4.3. Network Structure

To boost the efficiency of the baseline model and cut down on computational resource consumption, this study carries out lightweight fine-tuning of the original autoencoder network structure [37,38,39].

Specifically, the number of channels in the network is halved. This adjustment is made based on the analysis of experimental results. Figure 5 shows the network structure during our training. Figure 5a is the structure of the baseline (MUFusion). Figure 5b is the proposed network structure.

Colored arrows and dashed arrows represent dense connections and skip connections, respectively. The red font represents the number of channels. The dotted arrows and colored arrows in the figure represent jumping connections and dense connections. H and W are the dimensions of the image, representing the height and width. 3×3 is the size of the convolution kernel.

In our final network structure, the encoder part is used to extract features at different scales from source images, enabling the capture of information from details to the overall scene. Subsequently, the decoder part utilizes these features to reconstruct the final fused image.

## 5. Experiments

In this section, the experimental settings will be introduced. Subsequently, qualitative and quantitative experiments are conducted to analysis the fusion performance of our proposed method and other existing fusion methods on the proposed dataset, EDMF. Additionally, ablation studies are performed to evaluate the impact of the proposed improvements.

### 5.1. Experimental Details

Considering the brightness difference among source images, this work chooses to transform training image blocks into the YCBCR color space and the Y component is utilized to train our model. The Y components of the two image blocks are concatenated as the input of the network. Finally, the fused Y component is restored with CB and CR to form the RGB image.

As we discussed in introduction part, the MFIF task suffers from a lack of training datasets. In this study, to obtain sufficient training data, the first 500 pairs of multi-focus images from the EDMF are selected and cut, and 106,250 image blocks with a size of 256×256 are obtained. The number of image pairs used for testing is the remaining 500 pairs.

Network parameters are updated using the Adam optimizer, with a learning rate of 0.0001, 10 epochs, and a batch size of 48. All related experiments are conducted on an NVIDIA GeForce RTX 3090Ti GPU.

### 5.2. Evaluation Metric

In scenarios with exposure differences, in order to comprehensively evaluate the effectiveness of the proposed benchmark experiment, this paper selects eight metrics. The fusion results are comprehensively considered mainly from three aspects, including the degree of information preservation in the fused image, the degree of detail and structural information preservation in the fused image, and the information fidelity based on human vision. Considering these eight metrics in combination can reflect the richness of the detailed structural information and the degree of information preservation of the fused image in scenarios with exposure differences.

Specifically, these eight metrics are utilized: (1) Entropy (EN) to measure the information content of the fused image; (2) Mutual Information (MI) [40] to evaluate the amount of information retained between the source images and the fused result; (3) Spatial Frequency (SF) [41] to characterize the overall activity level of image details and textures; (4) Visual Information Fidelity (VIF) [42] to assess information preservation based on a human visual system model; (5) Objective Image Fusion Performance Measure (Qabf) [43] focusing on the preservation of edge strength and orientation; (6) Structural Similarity Index (SSIM) [44] to measure structural information retention; (7) Average Gradient (AG) reflecting the clarity of the image; and (8) Peak Signal-to-Noise Ratio (PSNR) as a traditional metric for estimating noise level in images.

These metrics form a comprehensive evaluation framework that allows us to accurately quantify the performance of fusion algorithms in dealing with exposure variations in real-world scenarios.

### 5.3. Comparative Experiments

This study compares seven representative MFIF methods: the pioneering work that introduced neural networks to the MFIF task (CNN) [26], end-to-end learning methods (GACN) [19], methods based on small area perception (SAMF) [17], zero-shot multi-focus image fusion (ZMFF) [20], fusion methods based on multi-dictionary linear sparse representation and region fusion model (MDLSR) [21], a generic fusion method using memory unit structures (MUFusion) [32], and unsupervised end-to-end generic fusion methods (U2Fusion) [31].

All these methods are implemented based on public code (released by the original authors) and the pre-trained parameters.

#### 5.3.1. Objective Analysis

As can be seen from Table 2, considering all the metrics comprehensively, the method proposed in this paper has the best performance. Specifically, it ranks second in the image fusion indicators EN, AG, and PSNR, which indicates that in terms of entropy, average gradient, and peak signal-to-noise ratio, although the method in this paper does not reach the optimal level, it also shows a relatively high level and is close to the optimal result.

The metric SF indicates that in terms of spatial frequency, the proposed method has a significant advantage and can better retain the spatial detail information of the image, so that the fused image has a more excellent performance in spatial features. It also reflects from the side that the method in this paper effectively utilizes and optimizes the spatial information when dealing with image fusion.

The performance on other metrics does not reach the best. This is attributed to the difference in the generation method of fused images. As mentioned in the related work in Section 2, for MFIF methods based on decision maps, the fused image is directly obtained by multiplying the source images by the corresponding elements of the map respectively and then adding them. These methods can retain the original information of the source images to the greatest extent. However, there are exposure differences in the EDMF dataset proposed in this paper. Although traditional methods perform excellently in most metrics, there are a large number of artifacts in the visualization results. The current evaluation metrics for image fusion have certain limitations, and further research on the evaluation of image quality is required.

Therefore, in the field of image fusion, especially for multi-focus image fusion, it is not suitable to only focus on metrics. The quantitative metrics may not show a significant advantage of our proposed method over the existing methods. It is crucial to consider the qualitative aspects of image fusion, particularly in terms of visual clarity and detail preservation.

#### 5.3.2. Subjective Analysis

The unsupervised fusion model proposed in this paper is visually compared with seven other fusion methods. As shown in Figure 6, Figure 7, Figure 8 and Figure 9, $I_{1}$ and $I_{2}$ are two source images.

The results indicate that the majority of MFIF methods generate fused images with numerous artifacts, primarily attributed to their heavy dependence on precise decision maps. This is also reflected in the objective indicators. From the PSNR values in Table 2, it can be clearly seen that the fused images obtained by the methods based on the decision map have more noise. Although both MUFusion [32] and U2Fusion [31] methods do not rely on decision maps and show relatively good visual effects, they still exhibit insufficient focus when handling details in the fusion images.

To highlight the differences in results between our method and the baseline (MUFusion), Figure 10 provides more examples of subjective comparisons. The fused images obtained by the training strategy refined in this paper are significantly better than MUFusion in terms of image texture details and clarity. In contrast, the proposed method demonstrated outstanding performance in qualitative assessment, indicating its superiority in handling complex exposure difference and maintaining image details. The fact that the SF value in Table 2 is much higher than that of MUFusion confirms this point.

In addition, qualitative experiments on commonly used datasets MFFW and Lytro are also conducted. As illustrated in Figure 11 and Figure 12, due to the lack of attention to exposure variation issues in public datasets, the fused images produced by other fusion methods exhibit no discernible visual difference.

### 5.4. Ablation Experiments

#### 5.4.1. Ablation Study About Adaptive Memory Unit

To validate the effectiveness of the proposed adaptive memory unit module and the spatial frequency-based activity level map, corresponding ablation experiments are conducted separately, as shown in Figure 13.

Figure 13a illustrates the fusion result of the baseline. Figure 13b presents the outcome after optimizing the memory unit into an adaptive memory unit. Figure 13c demonstrates the result obtained by employing spatial frequency for extracting active levels, instead of utilizing the original VGG feature extraction as active level weights. Figure 13d shows the final improved fusion result of this paper. From the visual results, the improved module proposed in this paper enhances the fused image, resulting in increased clarity and overall image quality.

As shown in Table 3, the adaptive memory unit structure enhances the retention of information from the source images. The introduction of spatial frequency improves the edge strength and information content of the fused image. While each individual improvement may not significantly impact performance, their combined application produces a complementary effect, demonstrating the overall effectiveness and superiority of the proposed method.

#### 5.4.2. Ablation Study About Adaptive Loss Ratios

To verify the rationality and scientific nature of the adaptive loss rates *W* and $W_{i}$, this paper conducted three groups of control experiments. In the first group, no weights are set. In the second group, fixed weights are adopted, with *W*, $W_{1}$, $W_{2}$, $W_{3}$, and $W_{4}$ all set to 0.2. In the third group, adaptive loss weights are used, which are adaptively adjusted during the training process.

The comparison of the fusion results are shown in Figure 14. The adaptive loss weights assign different loss ratios according to the quality of the fusion results obtained from different epochs stored in the memory unit during the training process. It is more scientific compared to not setting loss weights and using fixed loss weights. This is also reflected in Figure 14. A scientific setting of the loss weights is more conducive to the network learning more accurate network parameters and ultimately obtaining fusion images of better quality.

## 6. Conclusions

This paper addresses the issue of exposure difference in MFIF by introducing a new dataset and a new baseline. Public datasets are overly idealized and do not adequately reflect the common exposure differences present in real-world photography. Consequently, deep learning methods developed on these datasets often underperform in practical applications. The proposed dataset, EDMF, includes images captured with smartphones under various lighting conditions, covering diverse scenes without any adjustments to camera parameters, thereby realistically simulating real-world usage. We hope that the EDMF dataset will attract interest from researchers, fostering the development of more efficient and robust MFIF methods to better address exposure difference in practical applications.

Furthermore, although multi-focus image fusion in complex real-world scenes is now achievable, the method proposed in this paper still requires pre-registered images as input. Our goal is to develop a method that can immediately produce a fused image from multiple shots taken on the spot. Currently, there are mainly two solutions. The first one is to effectively utilize transfer learning and combine it with the fusion methods. The second is to construct an end-to-end unregistered image fusion framework. At present, we are still in the process of exploration.

## Figures and Tables

**Figure 1 sensors-24-07287-f001:**
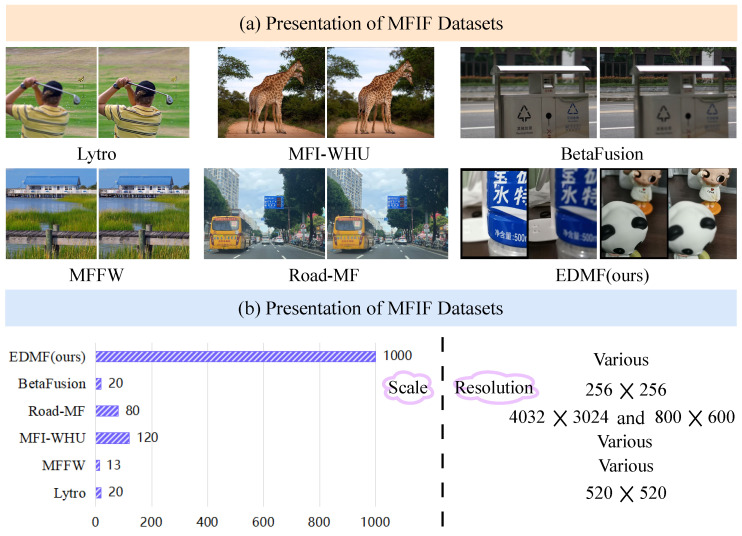
(**a**) Compared to the existing public datasets, the EDMF dataset not only has focus differences but also includes exposure differences. (**b**) The dataset disclosed in this paper has advantages in its scale and varying resolutions. (**c**) Taking the SAMF [17], GACN [19], ZMFF [20], MDLSR [21], and SESF [22] algorithms as an example, it is demonstrated that existing MFIF methods face the challenge of exposure difference.

**Figure 2 sensors-24-07287-f002:**
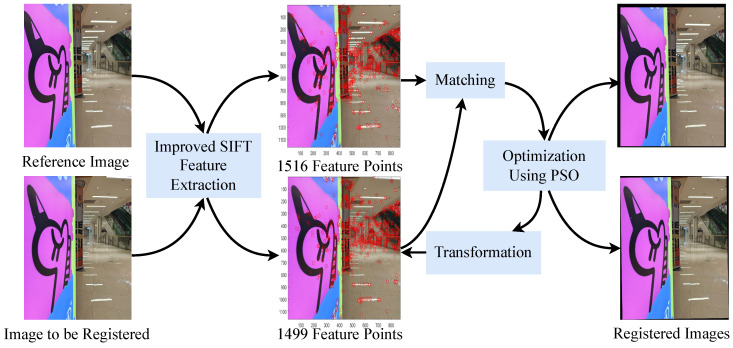
Registration process. During registration, one of the original images is taken as the reference for registration. The registration process includes four steps: feature point extraction, feature point matching, transformation matrix calculation, and source image resampling. The particle swarm optimization (PSO) algorithm improves the registration efficiency.

**Figure 3 sensors-24-07287-f003:**
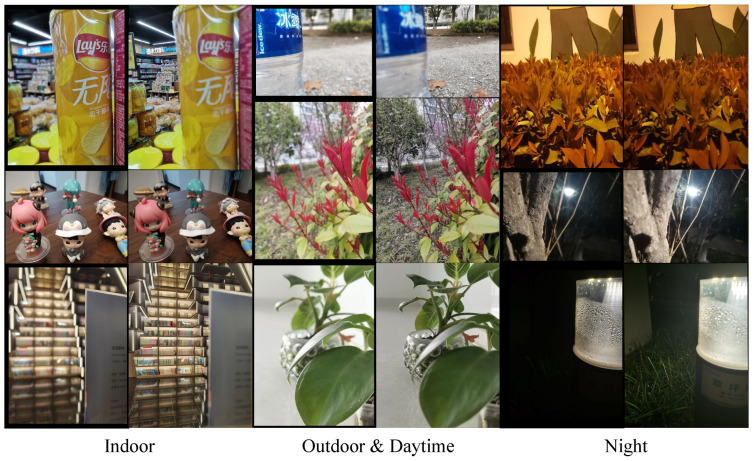
Partial image display of the EDMF dataset. It involves a variety of complex real-world scenes, including indoor, outdoor, day and night.

**Figure 4 sensors-24-07287-f004:**
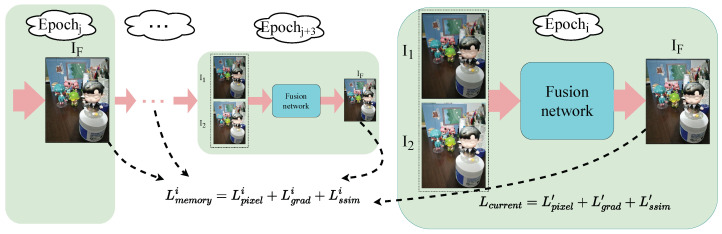
The proposed adaptive memory unit structure. Epoch_*i*_ represents the current training epoch. Epoch_*j*_ to Epoch_*j*+3_ represents the four epochs with the best performance in previous training epochs.

**Figure 5 sensors-24-07287-f005:**
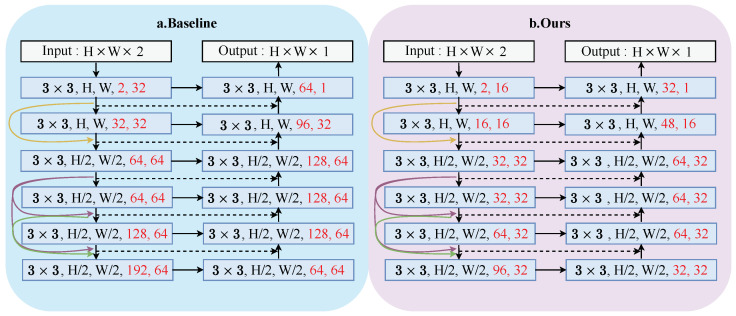
Network structure during training. (**a**) The network structure of the baseline (MUFusion). (**b**) The network structure of this paper. The red font represents the number of channels.

**Figure 6 sensors-24-07287-f006:**
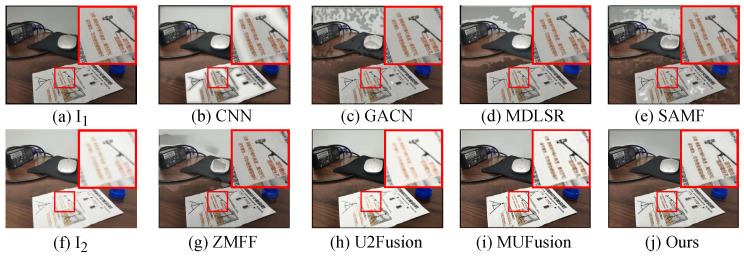
Qualitative visual comparison between our method and other representative methods on the No. 108 pair in EDMF dataset.

**Figure 7 sensors-24-07287-f007:**
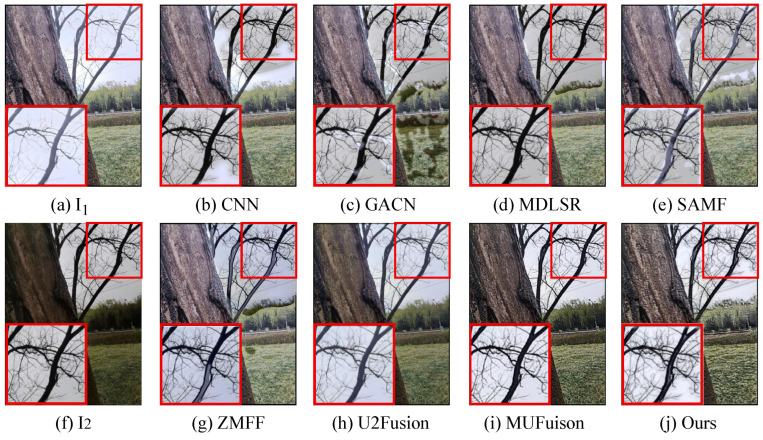
Qualitative visual comparison between our method and other representative methods on the No. 285 pair in EDMF dataset.

**Figure 8 sensors-24-07287-f008:**
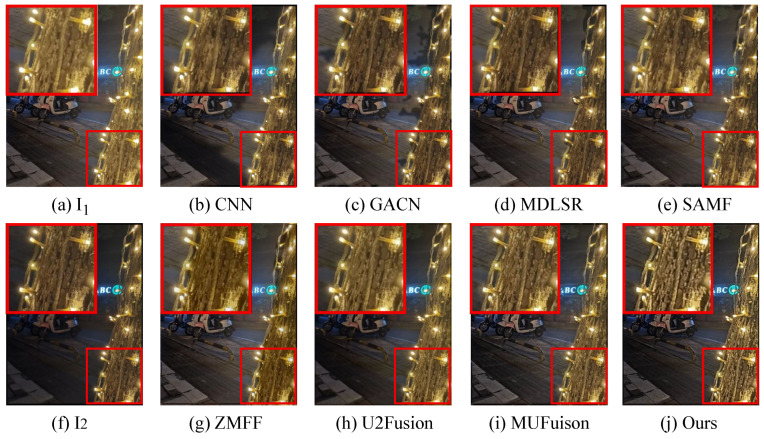
Qualitative visual comparison between our method and other representative methods on the No. 94 pair in EDMF dataset.

**Figure 9 sensors-24-07287-f009:**
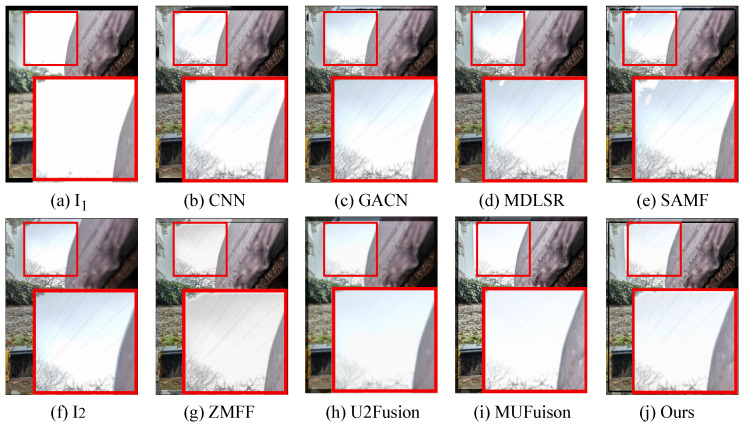
Qualitative visual comparison between our method and other representative methods on the No. 467 pair in EDMF dataset.

**Figure 10 sensors-24-07287-f010:**
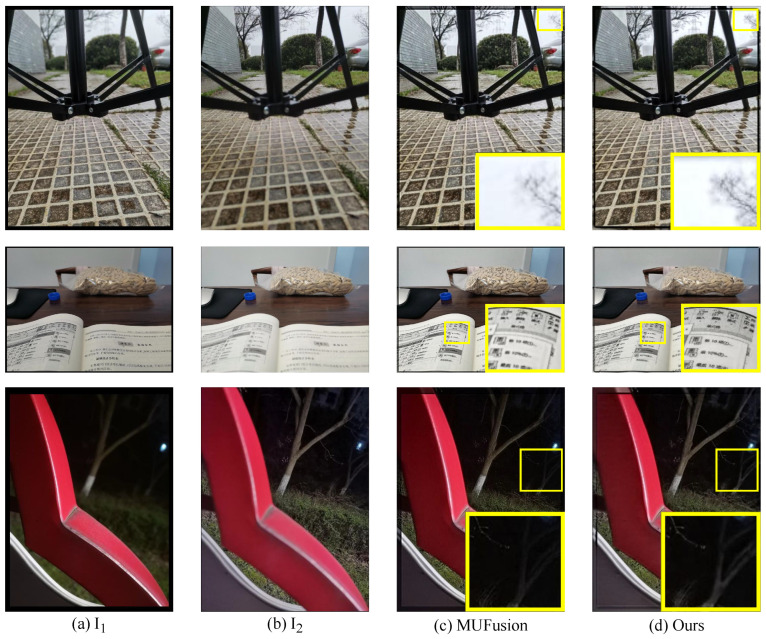
Subjective comparison of our fusion results with the baseline (MUFusion). The yellow boxes are used to visually highlight the significant differences in fusion results between our method and MUFusion.

**Figure 11 sensors-24-07287-f011:**
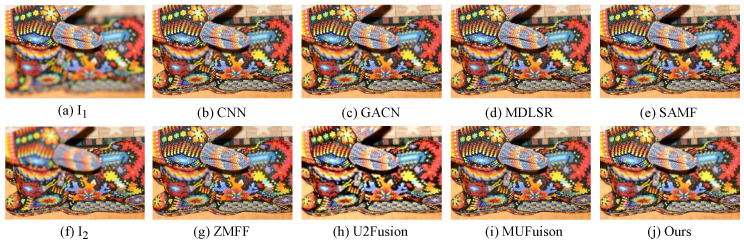
Qualitative visual comparison on MFFW dataset between our method and other representative methods.

**Figure 12 sensors-24-07287-f012:**
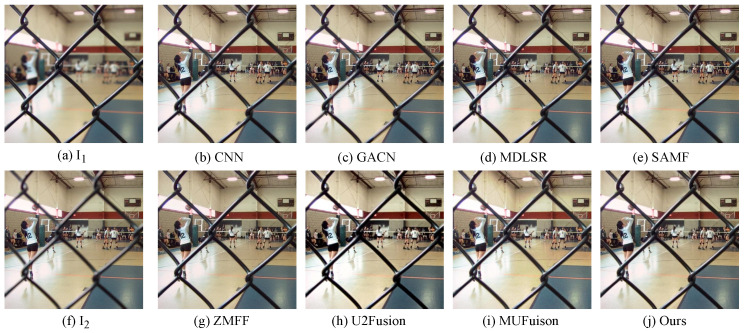
Qualitative visual comparison on Lytro dataset between our method and other representative methods.

**Figure 13 sensors-24-07287-f013:**
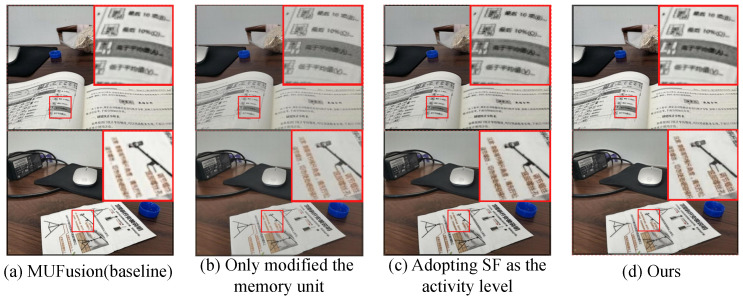
Qualitative visual comparison between our method and other representative methods.

**Figure 14 sensors-24-07287-f014:**
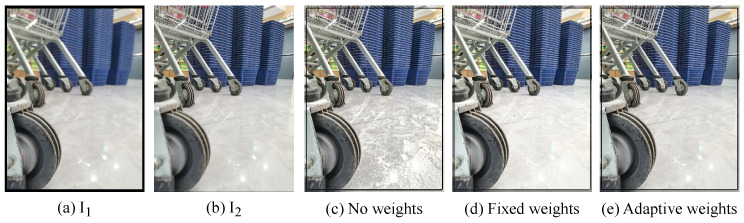
The impact of different loss ratio settings on the fusion results.

**Table 1 sensors-24-07287-t001:** The EDMF and existing datasets are compared respectively from four aspects: data generation method, number and size of images in the dataset, whether it is a real shot, and whether there is an exiting exposure difference.

Dataset	Data Generation Method	Size	Realistic	Exposure
Lytro	Captured by light field camera	20 pairs 520 × 520	Yes	No
MFFW	Online collection	13 pairs Various	Yes	No
MFI-WHU	Synthetically generated	120 pairs Various	No	No
Road-MF	Captured by light field camera	80 pairs Various	Yes	No
BetaFusion	Synthetically generated	20 pairs 256 × 256	No	No
EDMF(ours)	Captured by mobile phone	1000 pairs Various	Yes	Yes

**Table 2 sensors-24-07287-t002:** Comparison of objective metrics on proposed datasets. **Bold**: Best, Red: Second Best, Blue: Third Best.

Methods	EN	MI	SF	VIF	Qabf	SSIM	AG	PSNR
CNN (2017)	7.484	7.879	19.716	1.347	0.686	**0.683**	6.135	67.963
GACN (2022)	7.522	7.468	20.632	1.288	**0.706**	0.645	6.686	68.156
MDLSR (2024)	7.447	**8.298**	20.009	**1.362**	0.699	0.680	6.309	67.780
SAMF (2024)	7.528	7.543	20.445	1.272	0.685	0.639	6.588	68.184
ZMFF (2023)	7.499	5.463	21.328	1.069	0.652	0.645	6.997	67.781
U2Fusion (2022)	7.479	6.160	12.135	1.073	0.534	0.682	4.575	**69.888**
MUFusion (2023)	**7.613**	5.324	20.247	1.067	0.563	0.640	**7.326**	68.545
Ours	7.567	5.598	**21.498**	1.189	0.653	0.667	7.121	68.720

**Table 3 sensors-24-07287-t003:** Metrics comparison of ablation experiments. **Bold**: best.

Methods	EN	MI	SF	VIF	Qabf	SSIM	AG	PSNR
MUFusion (baseline)	7.613	5.324	20.247	1.067	0.563	0.640	**7.326**	68.545
Only modified memory unit	7.489	**6.474**	14.791	1.171	0.623	0.661	4.145	**69.870**
SF as the activity level	7.562	5.679	17.303	1.181	0.617	0.653	5.339	68.731
Ours	**7.567**	5.598	**21.498**	**1.189**	**0.653**	**0.667**	7.121	68.720

## Data Availability

The dataset and other related data of this work are available at https://github.com/stywmy/EDMF (accessed on 10 November 2024).

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
