# Peer review of "EDMF: A New Benchmark for Multi-Focus Images with the Challenge of Exposure Difference"

_sensors, 2024, doi:10.3390/s24227287_

Round 1
Reviewer 1 Report
Comments and Suggestions for Authors
1. Is the comparison method shown in Figure 1 a result of retraining or directly using the trained model for output? If the comparison method is supervised learning, it need to be retrained.
2. In Figure 5, compared with MUFusion, the feature count has been reduced by half. From what perspective is this design considered and whether increasing the feature count could yield better results?
3. Have the authors conducted specific experimental comparisons on public datasets other than the EDMF dataset to validate the generalization ability of the proposed method?
4. How do the authors formally describe and quantify the concept of ‘exposure difference’ in the multi focus image fusion method proposed in the paper? In addition, what are the theoretical innovations of the proposed unsupervised learning method based on memory units? Can you elaborate on the specific improvements in mathematical model or algorithm complexity compared to existing technologies when processing images with significant exposure differences using this method?
5. The title of Figure 1 is incorrect, with an additional number added; Figure 5 suggests consistency with the context, using lowercase letters (a); The numbering order in Figure 10 is incorrect.
Reviewer 2 Report
Comments and Suggestions for Authors
In the 'Related Work' section, it is recommended to include more details on comparing with existing datasets when introducing them, in order to more clearly demonstrate the advantages of the EDMF dataset in handling exposure differences.
When describing the baseline method in the article, detailed implementations and principle introductions of different modules should be added to help readers better understand how each module handles exposure differences in complex scenes.
In the experimental section, it is suggested to supplement the significance of quantitative evaluation indicators further, explain why these indicators were chosen, especially in scenarios with exposure differences, and how these indicators reflect the advantages and disadvantages of the algorithm.
In the " Conclusion, " it is suggested that further exploration of future work be added, especially on how to improve the model's adaptability to unregistered images, to enhance the article's forward-looking and guiding significance.
Finally, TarDAL[1], SegMiF[2], and CoCoNet[3] are applied to the field of image fusion from different perspectives. You should include descriptions of these articles in the relevant work to increase the comprehensiveness of the article.
[1]Multi-interactive feature learning and a full-time multi-modality benchmark for image fusion and segmentation
[2]Coconet: Coupled contrastive learning network with multi-level feature ensemble for multi-modality image fusion
Reviewer 3 Report
Comments and Suggestions for Authors
This paper presents a valuable contribution to the field by introducing a novel dataset and an improved baseline for multi-focus image fusion under exposure variation. However, the paper would benefit from additional clarity, more detailed comparisons, and expanded discussions about its limitations.
Concerns:
-
Dataset Licensing and Reuse:
Although the dataset is available on GitHub, the paper does not discuss licensing or usage restrictions in enough detail to ensure proper reuse and benchmarking. Clarifying these aspects will increase transparency and encourage the adoption of the dataset by other researchers. -
Method Comparison and Related Work:
-
While the results show that the proposed method outperforms others, key details are missing:
- What specific challenges did existing methods face when dealing with exposure differences?
- Are there more recent or state-of-the-art approaches that should be included in the comparison to provide a broader context?
-
The related work section overlooks some relevant studies. I recommend citing and comparing your method with the following papers to strengthen the discussion:
-
Fusion of Infrared and Visible Images Using a Deep Learning Framework:
This paper offers insights into fusing heterogeneous data (infrared and visible), which parallels the challenge of handling exposure differences in multi-focus images. Comparing these approaches will help illustrate how your method addresses similarly complex fusion tasks.- DOI: 10.32604/cmc.2022.021125
-
A New Multi-Focus Image Fusion Method Based on Deep Learning:
This work introduces a recent deep learning-based fusion technique that could be useful for benchmarking. Including a comparison with it will better position your model within the landscape of multi-focus fusion methods.- DOI: 10.1016/j.jvcir.2022.103485
-
-
-
Loss Function Discussion:
- The discussion around the loss function (combining pixel, gradient, and SSIM loss) could be more rigorous with clearer mathematical derivations.
- The adaptive loss ratio should be better justified through visual examples or graphs that demonstrate its impact during training.
- Including convergence graphs would further enhance the clarity of your experimental results.
-
Limitations and Practical Considerations:
- Although the paper acknowledges certain limitations, a more detailed discussion is needed:
- The reliance on pre-registered images is a significant constraint that could limit the method’s practical application.
- Address the computational complexity and potential challenges for real-time deployment.
- Although the paper acknowledges certain limitations, a more detailed discussion is needed:
-
Repetition and Readability:
- Some parts of the introduction and experimental setup sections are repetitive. Streamlining these sections will improve readability and maintain the reader's engagement.
- The subjective analysis could be strengthened by incorporating user studies or clearer justifications for the observed qualitative differences.
There are grammatical errors and english of the manuscrpt needs prrofread
Round 2
Reviewer 1 Report
Comments and Suggestions for Authors
All my questions and comments are addressed in this version.